# U-to-C RNA editing by synthetic PPR-DYW proteins in bacteria and human culture cells

Mizuho Ichinose [1,2,3,4 ✉], Masuyo Kawabata[1], Yumi Akaiwa[1], Yasuka Shimajiri[1], Izumi Nakamura[1], Takayuki Tamai[1], Takahiro Nakamura [1,2], Yusuke Yagi[1] & Bernard Gutmann [1 ✉]

Programmable RNA editing offers significant therapeutic potential for a wide range of genetic diseases. Currently, several deaminase enzymes, including ADAR and APOBEC, can perform programmable adenosine-to-inosine or cytidine-to-uridine RNA correction. However, enzymes to perform guanosine-to-adenosine and uridine-to-cytidine (U-to-C) editing are still lacking to complete the set of transition reactions. It is believed that the DYW:KP proteins, specific to seedless plants, catalyze the U-to-C reactions in mitochondria and chloroplasts. In this study, we designed seven DYW:KP domains based on consensus sequences and fused them to a designer RNA-binding pentatricopeptide repeat (PPR) domain. We show that three of these PPR-DYW:KP proteins edit targeted uridine to cytidine in bacteria and human cells. In addition, we show that these proteins have a 5′ but not apparent 3′ preference for neighboring nucleotides. Our results establish the DYW:KP aminase domain as a potential candidate for the development of a U-to-C editing tool in human cells.

[1] EditForce, Inc., Fukuoka, Japan. [2] Faculty of Agriculture, Kyushu University, Fukuoka, Japan. [3] Center for Gene Research, Nagoya University, Nagoya, Japan. [4] Institute of Transformative Bio-Molecules, Nagoya University, Nagoya, Japan. ✉email: mizu-ich@editforce.jp; bernard@editforce.jp

Genome editing technology has revolutionized the study of human diseases and opened new opportunities to enhance human health. However, undesirable mutations in the genome can lead to lethality in clinical use. RNA editing offers a safer alternative to this limitation by synthesizing modified proteins without changing the genomic sequence. Mainly developed around CRISPR-Cas and ADAR, this technology enables the modification of adenosine (A) to inosine (I) or cytidine (C) to uridine (U)[1–5]. These technologies are based on a complex of a guide RNA pairing to a specific region on an RNA molecule, recruiting an enzyme to perform the editing activity. Recently, a new C-to-U RNA base-editor that does not require a guide RNA based on the pentatricopeptide repeat (PPR) technology was developed in plants[6].

PPR proteins are RNA-binding proteins localized in chloroplasts and/or mitochondria, and are composed of ~31–36 amino acid PPR motifs repeated in tandem[7]. In PPR proteins catalyzing the C-to-U transition, the RNA-binding domain consists of PLS triplets containing three types of PPR motifs (P1, L1, and S1). The last triplet, P2L2S2, differs in the amino acid composition. These editing factors end with two PPR or TPR-like motifs (E1 and E2) and a cytidine deaminase domain, called DYW, to follow the (P1L1S1)x-P2L2S2-E1E2-DYW arrangement.

The PPR domain provides the specificity to the target by following a PPR code, in which two amino acids within each PPR motif connect one nucleotide[8–11]. The functions of the L2, S2, and E1 motifs remain unknown despite few studies suggesting that these PPR-like motifs also participate in the sequence recognition by following a non-canonical PPR code where the E1 motif would bind the nucleotide at position −3[9,12].

Recently, two independent bioinformatics studies identified two subgroups of DYW domains[13,14]: the canonical DYW domain (called DYW:PG in this study), present in all land plants, and the DYW:KP domain, which is restricted to three land plant clades (hornworts, lycophytes, and ferns) in which U-to-C editing occurs in mitochondria and chloroplasts. The DYW:KP proteins also match well with the U-to-C editing sites according to the PPR code, making them good candidates for U-to-C editing factors[13]. Three regions are highly conserved in the DYW domain but differ in the DYW:PG and KP domains: the N-terminal region (called PG box), the putative active site (HxE(x)CxxC), and a C-terminal aspartate (D), tyrosine (Y), and tryptophan (W) amino acid triplet.

However, it is still unknown whether the DYW:KP proteins truly possess U-to-C RNA editing activity or C-to-U activity as the DYW:PG proteins. In this study, we successfully developed a U-to-C RNA editing factor based on DYW:KP proteins that is functional in bacteria and human cells.

## Results

### Engineering a U-to-C editing protein
Until recently, no method was available to transform hornwort or fern species, making the study of DYW:KP proteins in natural U-to-C hosts impossible. However, a synthetic PLS-DYW:PG protein based on consensus motifs has previously been used to successfully edit a targeted cytidine in plant chloroplasts[6]. With the confidence of this success, we designed seven DYW:KP proteins based on consensus sequences (Fig. 1).

Those proteins include an N-terminal PLS domain composed of three variants of the P1-L1-S1 triplet of PPR motifs. Each motif was designed to represent its location on the PPR array. For instance, the design of the first P1 motif is based on the most representative amino acids in the first P1 motif present in PLS proteins of 66 plant genomes (Fig. 1 and Supplementary Fig. 1).

This PPR-PLS domain was fused to seven different C-terminal domains, including the DYW domain and the five upstream PPR-like motifs (P2, L2, S2, E1, and E2) (Fig. 2 and Supplementary Fig. 2). Six of those domains were designed on small clades of DYW:KP proteins identified in phylogenetic trees constructed on the DYW proteins isolated from hornworts, lycophytes, and ferns transcriptomes[14] (Fig. 1 and Supplementary Fig. 3) to represent the diversity of DYW:KP domains in land plants.

In hornworts, two variants of DYW:KP proteins, named DRH and GRP after the conserved three last amino acids, form two distinct subgroups of proteins[13]. In this study, the amino acid sequences of designer KP1 and KP2 were mainly based on DRH and GRP proteins, respectively (Supplementary Fig. 3a). The GRP variant is under-represented in transcriptomes, indicating that the number and the diversity of sequences used to design the KP2 protein are low. To improve the designer GRP protein sequence, we designed a second protein, called KP3, based on GRP sequences isolated from the *Anthoceros angustus* genome[15] (Supplementary Fig. 3b).

The subclades of DYW:KP proteins from tracheophytes (lycophytes and ferns) clustered together and were distinct from hornwort proteins[14]. KP5, specific to lycophytes, and KP6, specific to ferns, were designed in the same subclade of DYW:KP proteins (Supplementary Fig. 3c, d). KP4, designed on another subclade of lycophyte DYW:KP domains, was the only designer protein composed of 136 amino acids, including the xHP amino acids missing in most of the DYW:KP proteins but highly conserved in DYW:PG (Fig. 2 and Supplementary Fig. 3c).

To complete the set of DYW:KP proteins, we designed a domain, termed KP7, based on the motifs isolated in the 133 amino acid DYW:KP C-terminal domains identified in the transcriptomes of land plants[14]. The amino acid sequence is closer to the tracheophyte designer DYW:KP proteins because of the abundance of fern species in the set of proteins used for the design (Fig. 2 and Supplementary Fig. 2).

A recent study shows that a designer DYW:PG protein could target specifically the chloroplast editing site on *rpoA* if the amino acids involved in the RNA recognition in L2, S2, and E1 motifs were replaced by the ones present in CLB19, the PPR protein targeting *rpoA* editing site in Arabidopsis[6]. We followed the same approach for this study (Fig. 1). The canonical PPR code was used for P1, L1, S1, and P2 motifs.

### Designer DYW:KP proteins have U-to-C editing activity
To assess whether the designer DYW:KP proteins could function as active editing factors, we first tested their activity in bacterial Rosetta2 *E. coli* cells, where both the CDS coding for the DYW:KP protein and the editing site corresponding to a cytidine or uridine are localized on the same mRNA molecule (Fig. 1). Among the seven designer DYW:KP proteins, three converted the targeted uridine to cytidine after 18 h of isopropyl β-D-1-thiogalactopyranoside (IPTG) induction in *E. coli*, while no C-to-U editing activity was detected (Fig. 3a, b). The two GRP proteins edited the target with a low efficiency of ca. 26% for KP2 and 22% for KP3, in contrast to KP6 which edited its target with an efficiency of approximately 50%.

The DYW:KP domain harbors the $HxE(x)_nCxxC$ signature specific to cytidine deaminase proteins. We generated catalytic mutants of DYW:KP proteins by introducing one glutamine-to-alanine (HAE → HAA) and two cysteine-to-alanine (CxxC → AxxA) point mutations in the active site (Fig. 3c and Supplementary Fig. 4). The HAA and AxxA mutations abolished the U-to-C editing activity, indicating that the DYW:KP domains indeed have a U-to-C editing activity. As the alanine in HxE (HAE) is specific to DYW:KP proteins, we also designed a DYW:KP mutant that replaced alanine with serine (HAE → HSE), which is well conserved in the canonical C-to-U DYW:PG domains (Fig. 3c). The editing efficiency in the mutant was abolished (KP3) or reduced to ca. 11% (KP2) and 7%

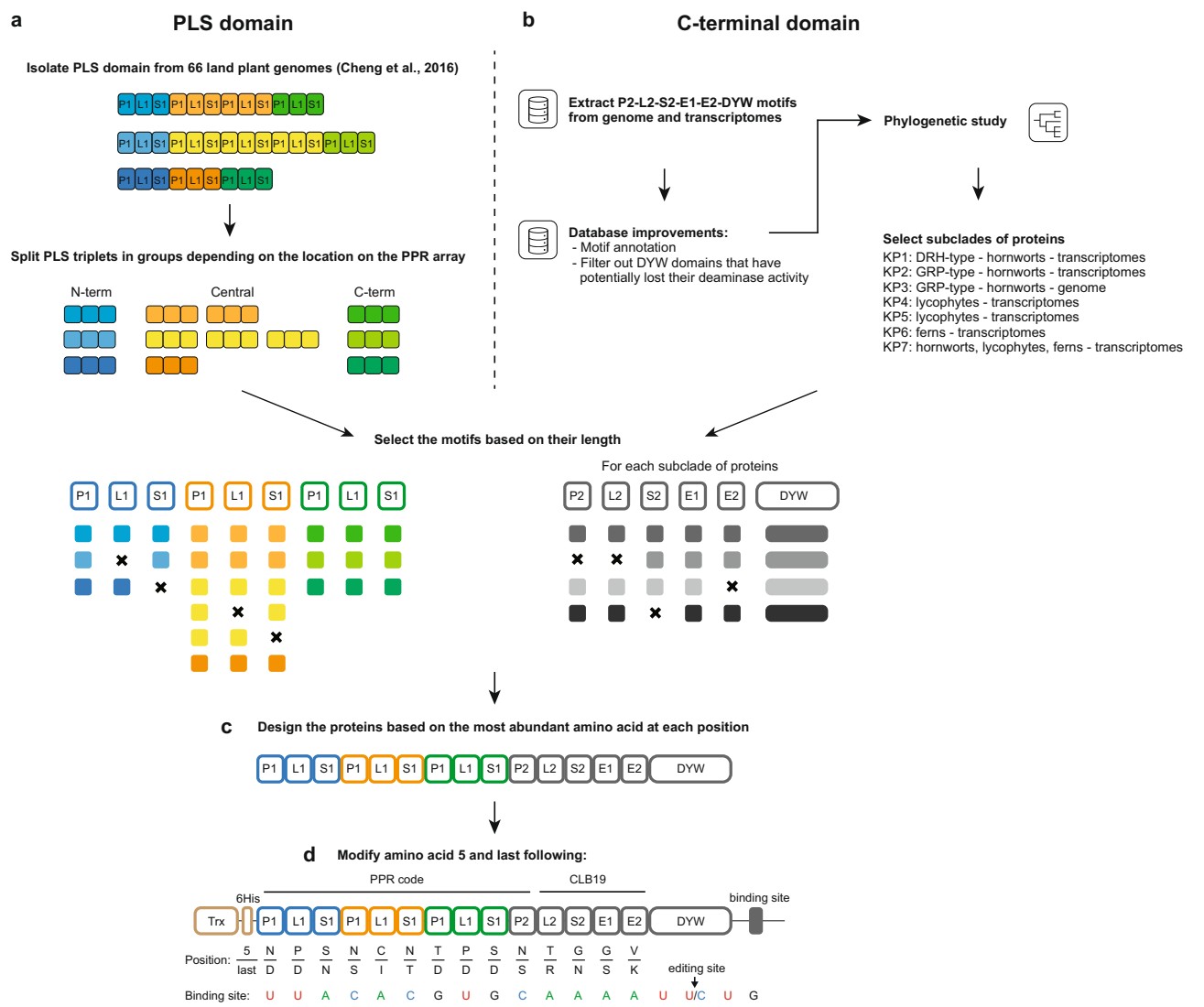

**Fig. 1 Schematic depiction of the workflow for designing the PLS and C-terminal domains of designer DYW:KP proteins. a** The P1L1S1 triplets used for the design of the PLS domain were isolated from land plant genomes and selected based on their location in the PLS domain: the 'N-term' triplets correspond to the first P1L1S1 triplet in the PLS domain, the 'Central' triplets are preceded and followed by a P1L1S1 triplet, and the 'C-term' triplets precede a P2L2S2 triplet. **b** The C-terminal domains were designed on the P2, L2, S2, E1, and E2 PPR-like motifs and DYW domains isolated in hornworts, lycophytes and ferns transcriptomes and *Anthoceros angustus* genome. After improving the motif database, a phylogenetic analysis was performed to isolate subclades of proteins. **c** After selecting the PPR motifs on their average length, a unique PLS domain composed of three P1L1S1 triplets and seven C-terminal domains (one for each subclade of proteins identified in (**b**)) were designed based on consensus sequences. **d** The amino acids involved in the RNA recognition were mutated to recognize specifically *AtrpoA*. The DYW:KP proteins overexpressed in Rosetta 2 cells were tagged with an N-terminal thioredoxin (Trx) domain and His-tag (6His). The targeted editing site (*AtrpoA*) is localized downstream of the stop codon. Below each PPR motif, the two amino acids determining the target specificity are aligned with the *AtrpoA* editing site. U/C indicates the editing site. Arrows indicate the design flow.

(KP6) indicating that this amino acid is important but not essential for the activity of the DYW:KP proteins.

**MORF2 and 9 do not improve the U-to-C editing activity.** The designer PLS domain based on angiosperm sequences requires multiple organellar RNA editing factor (MORF) proteins for efficient binding and RNA editing in bacteria[6,16]. Because the design of the PLS domain used in this study is mainly based on angiosperm sequences, we hypothesized that the editing activity of the DYW:KP proteins could be improved by co-expressing them with MORF proteins in *E. coli*. Based on the method of a previous study[6], we coexpressed DYW:KP proteins with MORF2 or MORF9. The editing efficiency did not increase in the presence of either MORF in any of the seven DYW:KP proteins, suggesting

that the C-terminal domain is the limiting factor for a high editing efficiency (Supplementary Fig. 5).

**KP6 has both C-to-U and U-to-C editing activities.** To test if the designer DYW:KP proteins have RNA editing activity in human cells, we cloned the gene coding for the DYW:KP proteins tested in bacteria, followed by their target site (*AtrpoA*) under the control of the cytomegalovirus (CMV) promoter. After overexpression of the protein for 24 h in human embryonic kidney 293T (HEK293T) cells, KP2, 3, and 6 proteins edited the targeted uridine with reduced editing efficiency compared to bacteria of ca. 11, 22, and 28% for KP2, 3, and 6, respectively (Fig. 4a, b). Surprisingly, KP6 protein showed not only U-to-C but also C-to-U editing activity in HEK293T cells, while no C-to-U editing

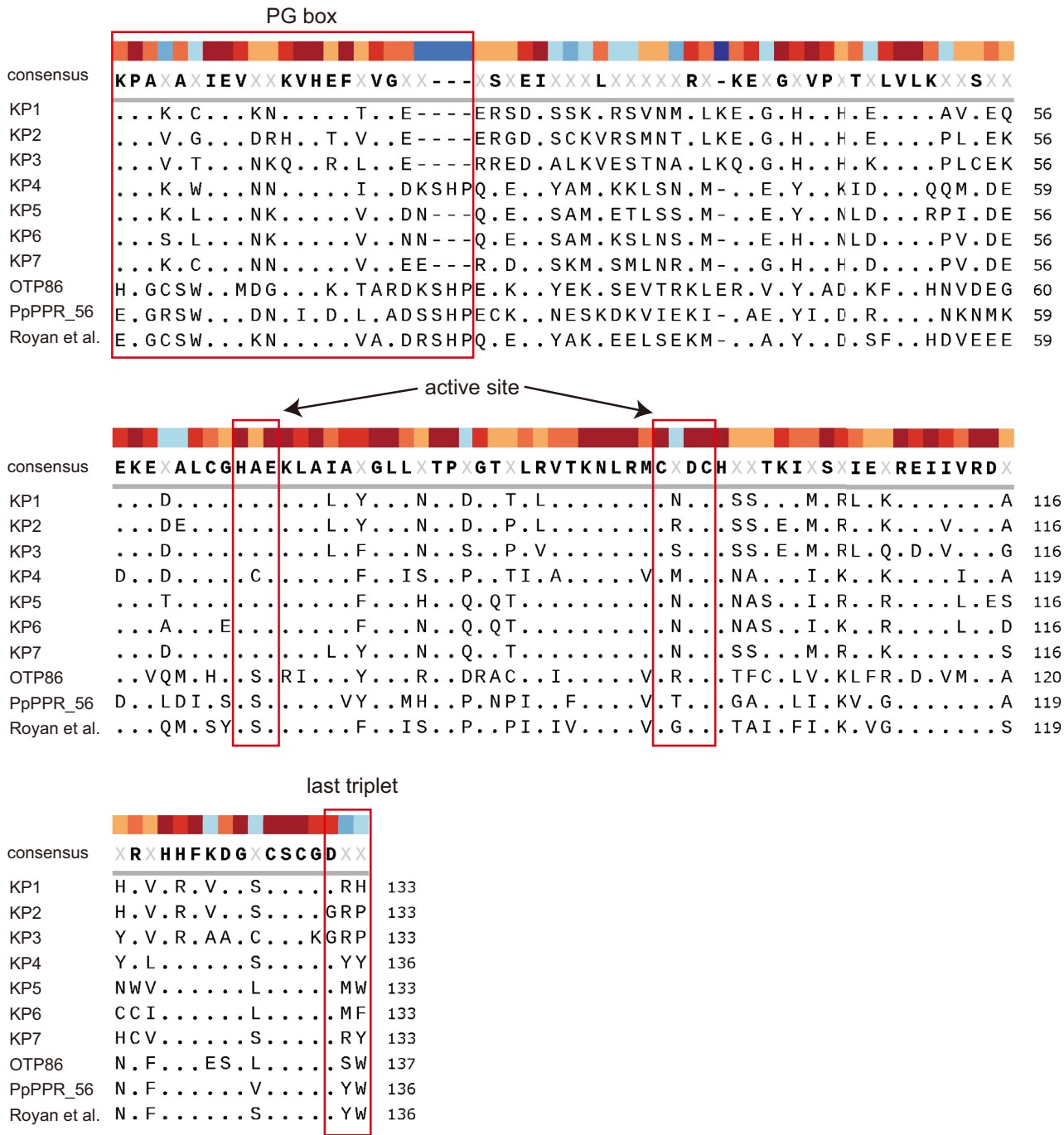

**Fig. 2 Comparison of the sequences of designer DYW:KP and DYW:PG domains.** The alignment of DYW domains compares seven designer DYW:KP domains with natural DYW:PG domains, OTP86[19,53], PpPPR_56[20,54], and a designer DYW:PG domain[6]. The alignment was prepared using MUSCLE[55]. The consensus sequence is shown at the top. Higher conservation is indicated by warm colors (such as brown and red) and lower conservation by cool colors (such as blue). Highly conserved regions in the DYW:PG domain are indicated.

activity was detected in KP2 or 3. As in *E. coli*, the catalytic KP2, 3, and 6 mutants (HAA and AxxA) exhibited no C-to-U and U-to-C editing activities in HEK293T cells. The mutation of alanine-to-serine (HAE → HSE) resulted in decreased C-to-U and U-to-C editing activities in KP6 (Fig. 4c), confirming that the same DYW domain can perform both C-to-U and U-to-C reactions. Interestingly, the editing efficiency of the KP2 HSE mutant increased by 10%, whereas no activity was detected with KP3 HSE confirming the important role of this amino acid in the function of the DYW domain.

**Designer DYW:KP proteins have a site preference.** Natural DYW domains are site-specific, and mutations around the editing site reduce or abolish the editing activity of the protein[17,18]. Thus, we suspected that the low editing efficiency of the KP proteins was driven by site-specificity, i.e., these proteins were not optimized for the *AtrpoA* editing site. We indirectly investigated the site preference of the designer DYW:KP2, 3, and 6 proteins at nucleotide positions −5 to +2 (relative to the edited nucleotide, 0) in HEK293T cells. The editing activity of each protein was analyzed by replacing the nucleotide at each position with one of

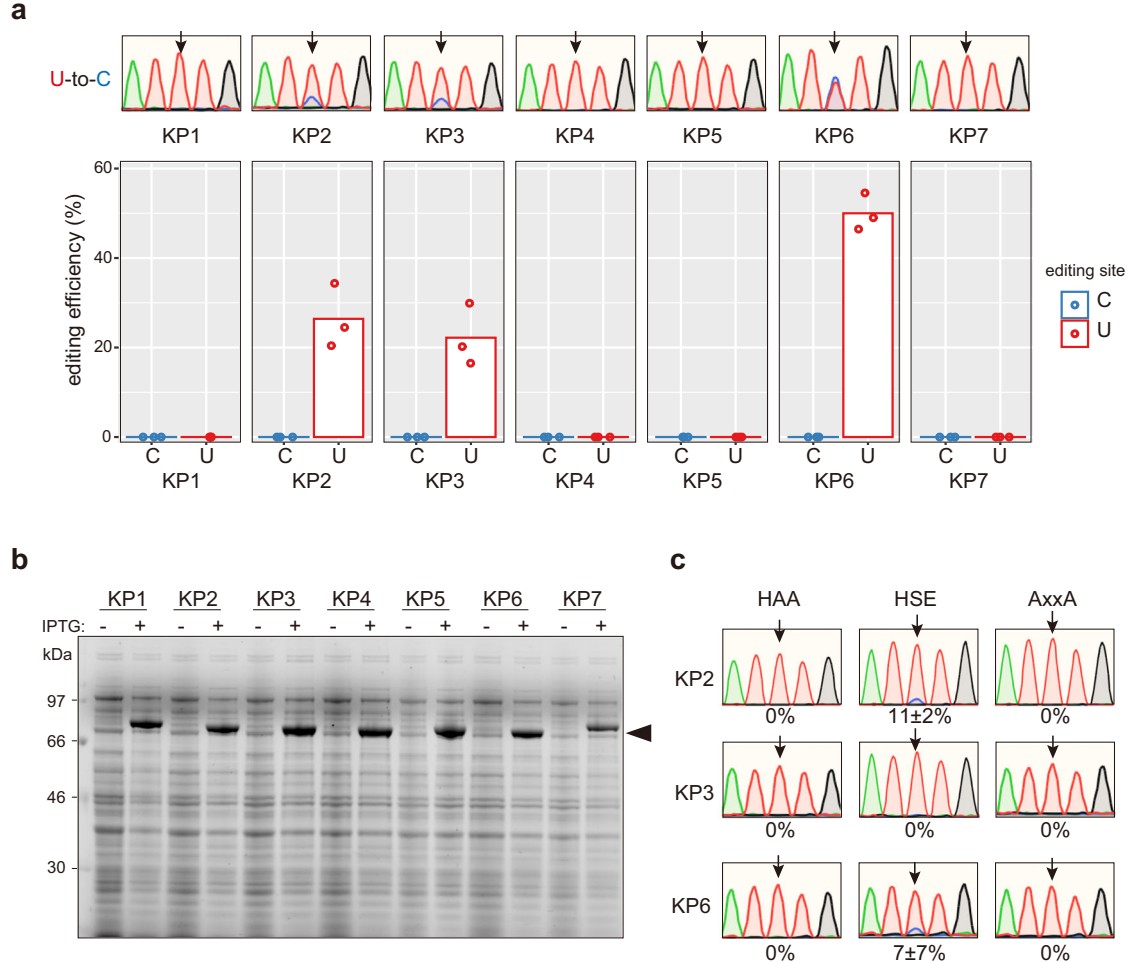

**Fig. 3 RNA editing activity of DYW:KP proteins in bacteria. a** The DYW:KP proteins exhibit U-to-C editing activity in bacteria. The sequence chromatogram corresponds to the cDNA sequence of the U-to-C editing sites. Arrows indicate the editing site. The editing efficiency is given as the mean of three biologically independent replicates. Each dot corresponds to one replicate. **b** The expression of the PPR proteins were verified on a denaturing sodium dodecyl sulfate-polyacrylamide gel electrophoresis (SDS-PAGE) gel after loading bacterial lysates before (−) and after (+) 18 h IPTG induction. Arrowhead indicates the predicted molecular weight of the DYW:KP proteins. **c** U-to-C editing relies on the DYW:KP domain. The editing efficiency of DYW:KP mutants is given as the mean ± s.d. of three biologically independent replicates.

the three other nucleotides (Fig. 5a–c). Theoretically, the editing efficiency of the DYW:KP protein should increase if the protein preference for the target increases. In contrast, the editing activity of the DYW:KP protein should decrease if the mutation in the nucleotide sequence affects protein binding.

As expected, the three DYW:KP proteins have a preference for A at positions −4 and −3, which are predicted to be recognized by the S2 and E1 motifs (Fig. 5a–c). In contrast, the preference of the L2 motif at position −5 varied depending on the DYW:KP protein. KP6 has a preference for A, contrary to KP2 and 3, suggesting that other amino acids in the PPR motif also modulate the specificity of the L2 motif to the nucleotide. The three DYW:KP proteins have a preference for purines and pyrimidines at positions −2 and −1, respectively, while less variability of the editing efficiency was observed when the nucleotides downstream of the editing site were mutated.

The nucleotide preference of KP6 at positions −1, +1, and +2 differs from the original target (Fig. 5c), with a significant preference for cytidine at position −1 and adenine and cytidine at position +1. To verify if KP6 improves the editing activity when the target sequence is optimized, we modified the nucleotide at positions −1 to +2. The U-to-C editing efficiency was increased

up to ca. 55% (Fig. 5d), while a small decrease in the C-to-U editing activity was observed.

Overall, those results suggest that the three DYW:KP proteins have a site preference in the vicinity of the editing site if we exclude that mutations lead to unpredictable variabilities in gene expression and/or structural modifications.

**Off-target editing by DYW:KP proteins**. The lack of editing of uridines in close proximity to the targeted editing site suggests that the DYW:KP proteins specifically recognize the target sequence they were designed to edit (Fig. 4b). Because no sequence similar to the binding site of the DYW:KP proteins was found on the plasmid mRNA, we performed transcriptome-wide RNA sequencing to detect potential C-to-U and U-to-C editing off-target sites for KP6, the designer DYW:KP protein with the highest editing efficiency. We retained the C-to-U and U-to-C editing sites that we considered appreciably edited (threshold of at least 5% editing) (Fig. 6a and Supplementary Data 1).

KP6 created 98 U-to-C editing off-target sites in HEK293T. The PPR protein performed 33% of U-to-C editing on *AtrpoA*, whereas the average editing efficiency on the off-target editing sites was 16%, suggesting that the relatively high editing efficiency

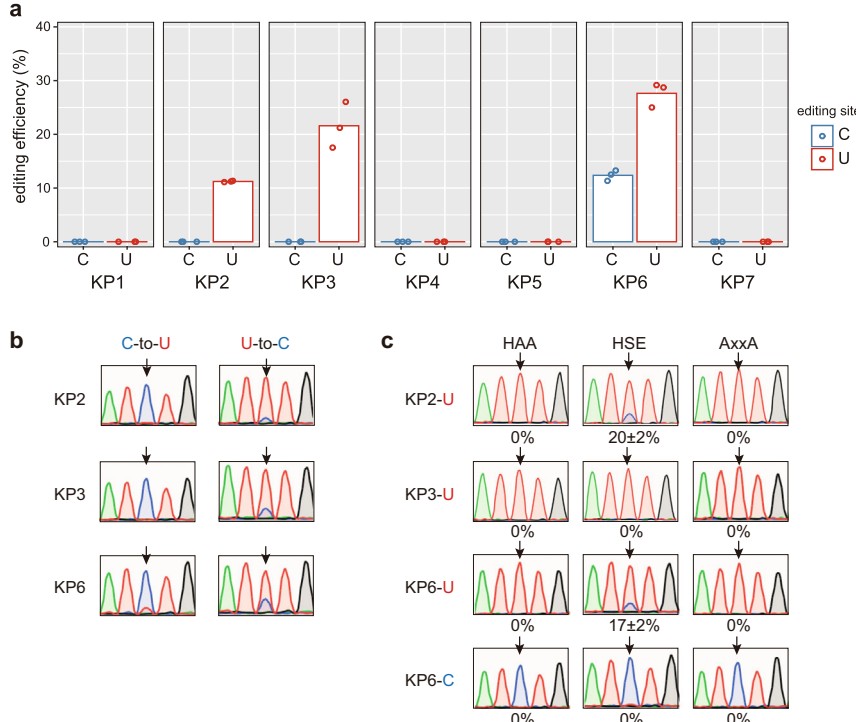

**Fig. 4 KP2 and 3 exert U-to-C editing activity, whereas KP6 has both C-to-U and U-to-C editing activities in human cells. a** The C-to-U (C) and U-to-C (U) editing efficiencies of seven DYW:KP proteins are given as the mean (bar) of three biologically independent replicates (dots). The chromatograms for one replicate are shown in (**b**). **c** The C-to-U and U-to-C editing activities depend on the DYW:KP domain. The editing efficiencies of DYW:KP mutants are given as the mean ± s.d. of three biologically independent replicates. The chromatogram of one replicate is provided as an example. Arrows indicate the editing site.

on *AtrpoA* is due to its location on the plasmid mRNA coding for the PPR protein. Among 255 potential off-target editing sites having less than four mismatches with *AtrpoA*, only three were edited (Supplementary Data 2). However, the sequence logo generated from an alignment of KP6 off-target sites shows a similarity of sequence with the *AtrpoA* editing site (Fig. 6b). The tolerance to mismatches increases in 5′end of the binding site and is considerably high for two nucleotide positions (−15 and −10) predicted to be recognized by the first P1 and second S1 motifs. The correlation between the PPR code chosen for the design of KP6 and the conserved nucleotide in the logo is strong apart from nucleotide −8 predicted to be recognized by the third L1 motif. Finally, the sequence logo suggests a nucleotide preference at position −1 but not downstream of the editing site, confirming the results obtained by point mutations (Fig. 5c).

Six significant C-to-U editing events were detected in KP6 transfected cells (Supplementary Data 1). The lack of similarity of five edited sequences to *AtrpoA* editing site suggests that those sites are edited independently of KP6. Nonetheless, one site edited at a frequency of 5% has high sequence identity with a U-to-C off-target consensus sequence suggesting that KP6 also has a C-to-U off-target editing activity (Supplementary Fig. 6).

## Discussion

To the best of our knowledge, no U-to-C editing factor had yet been characterized. Previous studies have demonstrated that DYW proteins are exclusive to C-to-U editing activity in seed plants. In this study, we reported that a subgroup of DYW proteins specific to seedless land plant clades has U-to-C editing activity.

The conserved differences between the functional designer DYW:KP proteins and the natural DYW:PG proteins were observed in the three essential regions: the PG box, the active site,

and the last triplet of amino acids. In DYW:PG, the S69 residue (in HSE(x)CxxC active site) plays an important inhibitory role in the cytidine deaminase activity of the DYW domain and a mutation (HSE to HAE) decreases the editing activity[19]. In DYW:KP proteins, this amino acid is a highly conserved alanine, and the reverse mutation (HAE to HSE) in KP2 protein increased the U-to-C editing activity in human cells while in KP6 protein reduced not only the U-to-C but also the C-to-U editing activity (Fig. 4), confirming that this position is important for the activity of the DYW domain, but does not discriminate between the two editing reactions. A mutation in D134 and W136 drastically decreased or abolished the editing activity in DYW:PG proteins[19,20]. Aspartate is highly conserved through the DYW:PG and KP proteins, apart from the GRP subgroup. In the DYW:PG domain, this amino acid consolidates the structure by forming a hydrogen bond with the highly conserved K103[19] that is replaced by glutamate in GRP proteins. Some other positions highly conserved in the DYW:PG domains are different in DYW:KP domains, including KP6; for example, V95 corresponds to M92 in DYW:KP domains. Based on these observations, we cannot conclude that the three essential boxes cited above are important for the binding of an amine substrate, the recognition of the targeted nucleotide, or the structure of the DYW:KP domain because of the reversible activity of the KP6 protein. However, this protein could represent a keystone between DYW:PG and DYW:KP to further understand the mechanism of the DYW domain.

Pioneering studies on the biochemistry of the C-to-U RNA editing activity in plant mitochondria and chloroplast suggested that this reaction operates through a hydrolytic deamination reaction and rejected the hypothesis of a RNA cleavage, transglycosylation and transamination mechanisms[21,22]. However, the irreversibility of the cytidine deaminase reaction under physiological conditions[23]

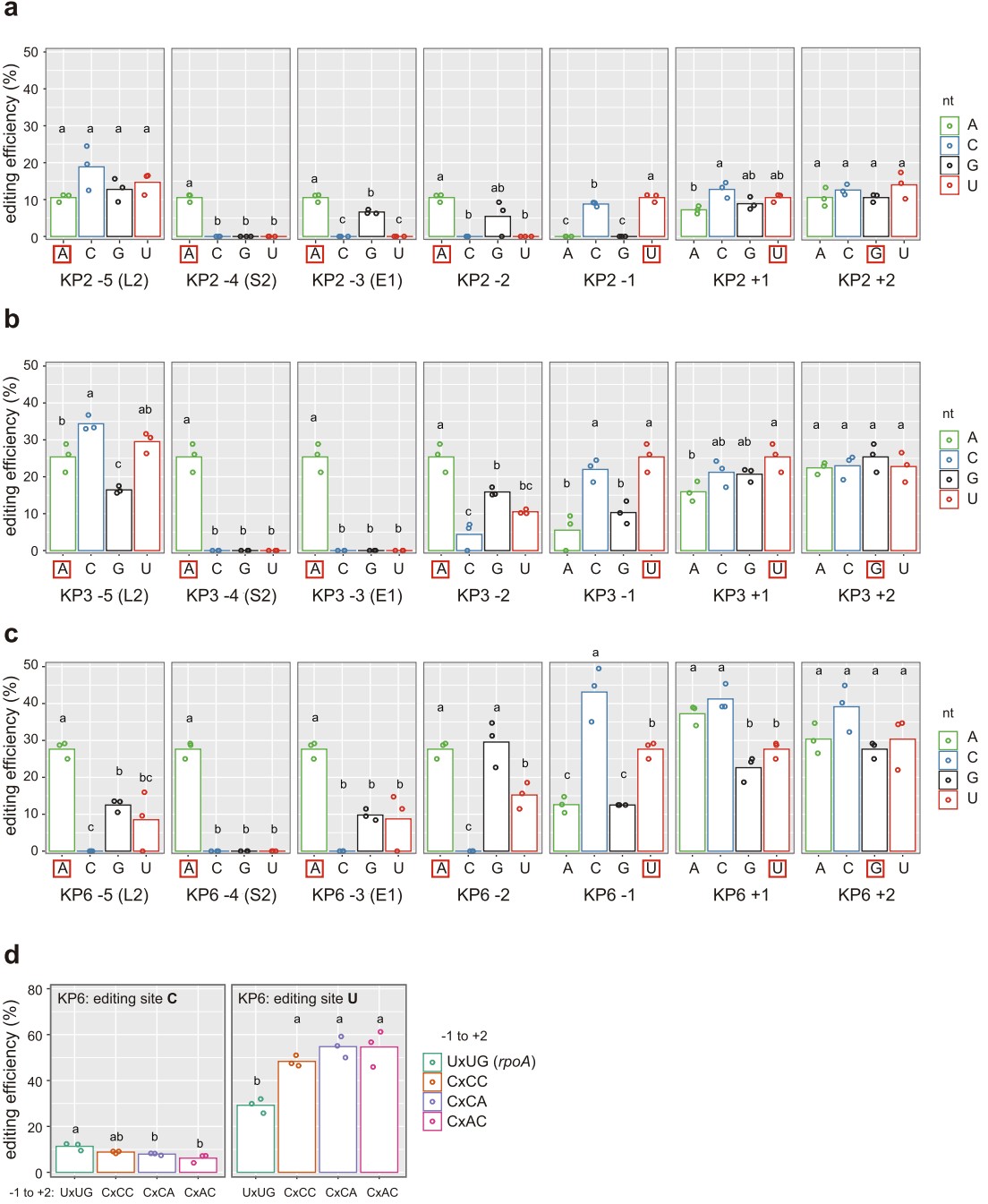

**Fig. 5 Designer DYW:KP proteins have a site preference.** Nucleotide preference of KP2 (**a**), KP3 (**b**), and KP6 (**c**) around the editing site in HEK293T cells. Positions −5, −4, and −3 are predicted to be recognized by the L2, S2, and E1 motifs. The color code indicates the nucleotides at each position. Red boxes correspond to nucleotides in the original target (*AtrpoA*). **d** The C-to-U and U-to-C editing efficiencies of KP6 when the target sequence at positions −1 to +2 is modified to follow the nucleotide preference are shown. Dots represent each replicate and bar, the mean of the editing efficiency of the three biologically independent replicates. Significant differences (one-way ANOVA, Tukey's comparison test, $P < 0.05$) are indicated with letters, multiple letters indicate that results are not significantly different for more than one group (e.g., **a** and **b** are significantly different while ab is not significantly different to **a** and **b**).

could not explain the U-to-C reaction in a same compartment of some plants, and the hypothesis of a transamination reaction catalyzing the C-to-U reaction was still investigated in vitro by testing potential acceptors of amino groups[24]. Recent studies identified distinct features suggesting that the catalytic mechanism of the DYW:PG domain differs from the other zinc-dependent nucleotide deaminases: a unique gating domain controlling the catalysis, a two-zinc ion per protein stoichiometry in the active form, and a failure

of a putative inhibitor to reduce the editing activity[19,25]. Therefore, we can hypothesize that the subtle differences in the amino acid sequence between the DYW:PG and KP domains could modify the catalytic pocket for the presence of an amine acceptor or donor to catalyze a unique cytidine deaminase or "reverse" reaction.

MORF2 and 9 are members of a 10-protein family and are essential cofactors for efficient editing in chloroplasts, including the *AtrpoA* editing site. Each C-to-U RNA editosome requires a

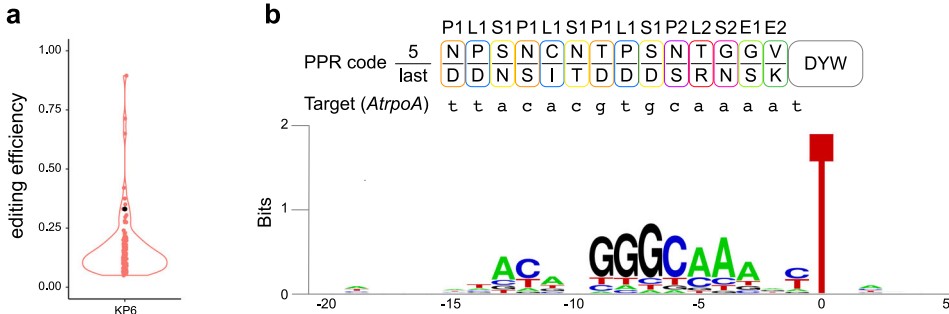

**Fig. 6 Off-target effect of KP6 in HEK293T cells. a** Violin plot shows the distribution of RNA editing efficiencies at the on (black dot) and off-target (pink dots) editing sites. **b** Base preference of KP6 around the editing site. The sequence logo was generated by Weblogo (http://weblogo.berlekey.edu/logo.cgi) from 98 U-to-C editing sites. Position 0 is the edited 'U'. PPR code combinations and on-target nucleotide sequence (*AtrpoA*) are shown above.

different protein combination, including a PPR protein and at least one MORF protein. In angiosperms, MORF proteins interact with the PLS triplets and E motifs to stabilize the PPR-RNA interaction and enhance the editing activity of the DYW:PG proteins[6,16,26]. The co-expression of MORF2 or 9 with a designer DYW:PG protein increased the editing efficiency by more than 20%[6]. Surprisingly, we could not obtain the same result with the designer DYW:KP proteins, while the PLS domains in the two studies had 95% identity, and the amino acids interacting with MORF proteins were identical in the second and third PLS triplets. It is possible that MORF2 and 9 proteins play a greater role by binding the C-terminal domain (P2-L2-S2-E1-E2) in the designer DYW:PG protein, or that the designer PLS domain used in this study requires a different MORF protein to improve the binding and editing activity. We suspect that factors other than MORF proteins could explain the low editing activity of the DYW:KP proteins: first, the speed of the DYW:KP protein to bind and be released depends on the specificity of the protein and the stability of the protein-RNA complex; second, the catalysis of two reversible transition reactions by the same domain could affect the efficiency of the U-to-C activity. The latter factor and the speed of both editing reactions could also partially explain the differences in editing activity observed between *E. coli* and mammalian cells. Indeed, the experimental conditions (e.g., sampling time or temperature) could have favored the editing activity of U-to-C over C-to-U in *E. coli*. For instance, in one experiment, we detected C-to-U editing activity in Rosetta2 after 30 h of induction but could not reproduce the results. Finally, a lower level of expression of DYW:KP protein can explain the low editing activity in human cells, where no DYW:KP proteins could be detected on SDS-PAGE or by western blotting.

Early works in angiosperms have identified the presence of a *cis*-element flanking the editing site from position −20 to +16[17,27–31]. They showed that the importance of the nucleotide and its position varies depending on the editing site, which is in line with the absence of a conserved consensus sequence for the C-to-U editing sites and the later discovery of the DYW proteins[17,30]. Our results are consistent with those studies and suggest that the designer DYW:KP proteins recognize specifically the sequence upstream of the editing site. In line with a recent study on DYW proteins[32], we showed that the editing efficiency of the designer DYW:KP proteins tends to decrease with a pyrimidine and a purine at positions −2 and −1, respectively (Fig. 5). The negative impact of a guanine at position −1 on the RNA editing efficiency was already highlighted for the C-to-U editing sites in angiosperms[17,30,31] and is similar to the neighbor preferences of other deaminases e.g., ADAR[33]. Surprisingly, the three designer DYW:KP proteins showed no preference to purines at positions +1 and +2, while some U-to-C editing events in early-diverging land plant lineages alter the premature stop codons.

After the discovery of a PPR code specific to the P1, P2, and S1 motifs, it was debated if the L1 motif works as a spacer between P1 and S1 motifs and/or contributes to the RNA recognition[34,35]. A recent study showed that recoding the L1 motifs of CREF3 protein affect strongly its RNA editing activity in plants suggesting that some of the L1 motifs can play a critical role in the RNA recognition mechanism[36]. The study of KP6 off-target editing sites shows a nucleotide bias to U and G at positions −14 and −8, respectively. Those two positions are recognized by an L1[PD] motif suggesting that the 5th and last amino acids are not sufficient to specify the nucleotide preference, and other amino acid positions in the L1 or adjacent PPR motifs should be taken into account in the future when retargeting the designer PPR proteins, e.g., position 2 sandwiching each nucleotide[10,37] or amino acids 9 and 13 interacting with the backbone phosphate/ribose groups of the nucleotide[37,38].

The development of programmable RNA editing tools, mainly based on ADAR, has been growing since the discovery of CRISPR-Cas but remains limited to C-to-U and A-to-I transition reactions[39]. The design of programmable PPR proteins has already been performed in plants and mammalian cells for the purpose of controlling the expression of transgenes and editing, stabilizing, and protecting RNA molecules[6,40–42]. Hence, with the characterization of the DYW:KP domain and the detection of U-to-C editing on human mRNA molecules, we increased the range of programmable editing enzymes, opening new opportunities for gene therapy. However, the low editing efficiency for the three DYW:KP proteins is a problem that will need to be solved to develop a biotechnological tool based on this study. The current designs of KP2, 3, and 6 have a site preference (e.g., pyrimidine at position −1), thus limiting the number of potential targets with a high editing efficiency. However, the diversity of editing sites in nature and the specificity of KP2, 3, and 6 proteins make us believe that we can reprogram the sequence recognized by the C-terminal domain. A second major improvement should target the PPR motifs. Although only a few works have studied the mode of recognition of the L motifs[34–36] the solution could come from the application of a PPR domain composed of a unique PPR motif repeated in tandem, for example, the SS motifs, a type of PPR motif upstream of the C-terminal domain, in some clades of seedless plants[7,43]. Despite the off-target effect on other mRNA molecules, DYW:KP proteins have a tremendous advantage over ADAR in terms of local off-target editing. Base editor tools based on ADAR produce frequent local off-target editing at the flanking adenine or cytidine[2,44]. In contrast, the DYW:KP proteins precisely edit the nucleotide four nucleotides downstream of the last PLS triplet (Figs. 3 and 4), without editing the two uridines directly flanking *AtrpoA*.

In conclusion, the characterization of the DYW:KP domain could be significant to further understand the mode of recognition

of the DYW domain and the reactions it catalyzes. This domain could also represent a valuable base editor to engineer a programmable U-to-C editing factor and complete the RNA toolbox.

## Methods

**Design of the PLS proteins**. The PLS domain composed of three triplets of P1, L1, and S1 PPR motifs was designed on the PPR motifs isolated from 66 land plant genomes (https://ppr.plantenergy.uwa.edu.au/ppr/)[7]. The motifs were selected based on their length (35 aa for P1 and L1 and 31 aa for S1 motifs) and the location on the PPR array in relation to 1/ the first P1L1S1 triplet based on the first P1, L1, and S1 motifs in PPR proteins starting with a P1 motif, 2/ the central P1L1S1 triplet on the P1, L1, and S1 preceded by at least three PPR motifs, followed by a P1L1S1 triplet, and 3/ the last P1L1S1 on the PPR motifs preceding the P2L2S2 triplet. The consensus sequences for each PPR motif were derived from EMBOSS cons[45] with a plurality value of 0. The number of motif sequences used for the design is shown in Supplementary Table 1.

To design the C-terminal domain, we first isolated DYW proteins from the *Anthoceros angustus* genome[15] (GCA_010909165.1) using the method described in refs. [7,14] and the hmm files from ref. [7] and ref. [14] for the PPR motifs and the DYW domain, respectively. The P2, L2, S2, E1, and E2 PPR-like motifs, as well as DYW domain were extracted from the *A. angustus* PPR dataset and the 1KP PPR database (https://ppr.plantenergy.uwa.edu.au/onekp/)[14], and the motif-size gaps were annotated according to the expected motif. The DYW proteins were filtered to remove proteins lacking the HxE(x)CxxC deaminase signature and the proteins displaying more than 34 unannotated amino acids upstream of a unique DYW or E2-DYW domain, to avoid including any proteins that have lost deaminase activity or DYW1/DYW2-like PPR proteins. Proteins with a DYW domain equal to or lower than 115 amino acids were removed from this study.

The partial or full-length C-terminal domains, including at least a DYW domain, were aligned using the L-INS-I mode in MAFFT (v7.407)[46]. Alignments were trimmed using TrimAL (v1.4.rev15)[47], with a minimum conservation threshold and a gap threshold of 20%. For phylogenetic reconstitution, we used FastTree (v2.1.10 Double precision)[48] with the WAG model of amino acid evolution[49] and discrete gamma model with 20 rate categories. For the selected clades of the DYW:KP proteins, the consensus sequences for each PPR motif were derived from the EMBOSS cons[45] with a plurality value of 0 after selecting motifs of equal length. The number of motif sequences used for the design of the PPR-like motifs and DYW domains is indicated in Supplementary Table 2.

**Cloning of PLS-DYW:KP proteins and targets for the editing assay in bacteria.** All primers used in this study are listed in Supplementary Table 3. Synthetic genes encoding the DYW:KP protein, including P2-L2-S2-E1-E2-DYW, were synthesized as gBlocks® Gene Fragments (Integrated DNA Technologies). The full-length PPR-PLS domain and the target sequence, including two *Bpi*I and *Esp*3I sites at the 5′- and 3′-ends, respectively, were synthesized by GENEWIZ. The gene was constructed in four sections (thioredoxin, PPR-PLS array, C-terminal domain, and target sequence including the editing site) using a two-step Golden Gate assembly. pET21b + PA was modified to remove the former *Esp*3I and *Bpi*I restriction sites and obtain a cloning site containing two *Esp*3I sites. Three parts were cloned into the modified pET21b vector by Golden Gate assembly using the *Esp*3I: 1/ Thioredoxin-6His-TEV cleavage site gene, including a *Bpi*I restriction site in the 3′-end, 2/ codon-optimized human C-terminal DYW:KP (P2L2S2E1E2DYW) gene, and a *Bpi*I restriction site in the 5′-end, and 3/ sequence coding for the editing site. The full-length PPR-PLS domain, including two *Bpi*I sites at the 5′- and 3′-ends, respectively, was cloned by Golden Gate assembly using *Bpi*I in the previously cloned plasmid. Point mutations were introduced into the KP domain by site-directed mutagenesis PCR.

**RNA editing in *E. coli***. To analyze the editing activity of the recombinant proteins in bacteria, we modified the protocol developed by[20]. Plasmids were introduced into Rosetta 2 (DE3) cells, and 2 mL *E. coli* starter cultures (LB with 100 μg mL⁻¹ ampicillin) were grown overnight at 37 °C. One hundred microliters of preculture were used to inoculate 5 mL LB with 100 μg mL⁻¹ ampicillin in a 24-deep well plate. The cultures were grown at 37 °C and 220 rpm until a representative well reached an $OD_{600}$ of 0.4–0.6. The plate was then cooled at 4 °C for 10–15 min before adding 0.4 mM $ZnSO_4$ and 0.4 mM IPTG. The plates were incubated at 16 °C and 220 rpm for 18 h. Two samples (500 μL) for RNA extraction and SDS-PAGE were centrifuged, and the resulting pellets were stored at −80 °C.

To stabilize bacterial RNA, RNAprotect Bacteria Reagent (Qiagen) was used according to the manufacturer's instructions. After adding 30 μL of lysozyme buffer (30 mM Tris-HCl; pH 8.0, 1 mM EDTA, 50 mg mL⁻¹ lysozyme) and 20 μL of proteinase K solution (TaKaRa) to a pellet, the RNA was extracted using the RNeasy Mini Kit (Qiagen) according to the manufacturer's instructions. RNA was subsequently treated with RQ1 RNase-Free DNase (Promega), and the cDNA was synthesized using ReverTra Ace® (TOYOBO) using 300 ng RNA and 1.25 μM random hexamer primers. The editing site was amplified using PrimeSTAR Max DNA® polymerase (TaKaRa), 1 μL of cDNA, and primers on the DYW domain and T7 terminator (Supplementary Table 3). PCR products were cleaned with ExoSAP-IT™ Express PCR Cleanup Reagent (ThermoFisher) and sequenced

(GENEWIZ) with forward primers specific to the DYW domain (Supplementary Table 3). RNA editing was analyzed from the raw sequencing chromatograms using EditR available at http://baseeditr.com/[50]. The ratio of U-to-C editing was quantified as the ratio of the percent cytidine area to the sum of the percent cytidine and thymidine areas, and vice versa for the C-to-U editing ratio. If the percent area was not significantly different from the noise after trimming, the given value for the percent area was 0 (*P*-value cutoff: 0.01). The experiments were repeated three times with independent primary clones.

**Cloning for the expression in mammalian cells**. The region encoding the 6His-PLS-DYW:KP protein and editing site was amplified from the plasmids prepared for RNA editing assay in *E. coli* with the respective primers to append two *Esp*3I sites at the 5′- and 3′-ends, respectively. The fragment was cloned into the mammalian expression vector PM18033 using Golden Gate Assembly with *Esp*3I, which carries the human cytomegalovirus (CMV) immediate-early promoter, the human β-globin chimeric intron, and SV40 polyadenylation signal.

**Mammalian cell culture**. HEK293T cells (RIKEN, RCB2202) were cultured in Dulbecco's Modified Eagle Medium (DMEM) with high glucose, glutamine, phenol-RED, sodium pyruvate (Wako), additionally supplemented with 10% fetal bovine serum (Capricorn) and 1% penicillin-streptomycin (Fujifilm) in a humidified $CO_2$ (5%) incubator at 37 °C. Cells were passaged every 2–3 days after reaching approximately 80–90% confluency.

**Transfections**. For the RNA editing assay, HEK293T cells were plated at a density of approximately $8.0 \times 10^4$ cells/well into 24-well flat-bottom cell culture plates (ThermoFisher) 24 h prior to transfection. For each well on the plate, 500 ng transfection plasmids were combined with Opti-MEM® I Reduced Serum Medium (ThermoFisher) and 1.5 μL of FuGENE® HD Transfection Reagent (Promega) to a total of 25 μL, and incubated at room temperature for 10 min, after which they were added to the cells. Cells were incubated at 37 °C and collected 24 h after transfection.

**RNA editing in HEK293T cells**. Cells were homogenized with 1-Thioglycerol/ Homogenization solution (supplied by Maxwell® RSC simplyRNA Tissue Kit; Promega) and Proteinase K, and RNA was extracted using the RNeasy Mini Kit (Qiagen), according to the manufacturer's instructions. DNase treatment, cDNA synthesis, RT-PCR, and direct sequencing were performed as described above.

**Off-target screening**. RNA extraction was performed with Maxwell® RSC simplyRNA Tissue Kit (Promega) and Proteinase K. Libraries were prepared by GENEWIZ from total RNA using poly(A) enrichment of the mRNA to remove ribosomal RNA. MGI tech DNBSEQ-G400 (2 × 150 bp), and Illumina Novaseq 6000 sequencing was performed by GENEWIZ for two and one replicate, respectively. Reads were mapped to the GRCh38.105 reference genome with STAR[51] (v2.9.7, parameters—quantMode TranscriptomeSAM—outFilterType BySJout—outFilterMultimapNmax 1—outSAMstrandField intronMotif). RNA editing candidate sites were analyzed with REDItools[52] (v1.2.1 parameters: -t 13 -e -d -l -U [AG,TC,CT,GA] -p -u -m 20 -T 6-0 -W -v 10 -n 0.05 -g 2 -s 1). Any significant sites identified in the transfected cells with the empty vector were considered as artifacts or SNPs. The editing sites were considered significant if the read count was above 10, the editing frequency different than 1 and the *p*-value calculated by Fisher's Exact Test was below 0.05 after correction for multiple tests using the Benjamini–Hochberg correction. The editing sites corresponding to known SNP positions were filtered with the NCBI SNP database (v151).

**Statistics and reproducibility**. Experiments were independently repeated three times. The results of the editing efficiency by direct sequencing are displayed either as the mean value ± standard deviation (s.d.) or a dot-plot graph, each dot representing a replicate and the bar the mean value. Significant differences ($P < 0.05$) between different groups were determined by the one-way ANOVA Tukey's comparison test and indicated with letters. The statistical calculations on the data were made using Rstudio.

**Reporting Summary**. Further information on research design is available in the Nature Research Reporting Summary linked to this article.

## Data availability

Uncropped and unedited images corresponding to the protein gels and blots are provided in Supplementary Figs. 7–11. The data used to generate the graphs are available in Supplementary Data 1 and 3. Sanger sequence chromatograms used to score editing efficiencies are available as Supplementary Data 4. Transcriptome wide RNA-sequencing data are available at Sequence Read Archive under BioProject accession number PRJNA856069. Expression plasmids are available from addgene under a uniform biological material transfer agreement (accession IDs: 190955 to 190994). All other data are available from the corresponding authors on reasonable request.

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

## Acknowledgements

We thank I. Small and M. Sugita for the valuable discussion at the initial stage. This work was supported by Research Grant for Young Scholars funded by Yamanashi Prefecture to M.I.

## Author contributions

M.I. and B.G. designed and coordinated the experiments. M.I., M.K., Y.A., Y.S., I.N., and B.G. performed experiments. M.I. and B.G. analyzed the results. M.I., T.T., Y.Y., and B.G.

performed the off-target analysis. M.I. and B.G. wrote the manuscript and designed the figures with support from Y.A., T.N., and Y.Y.

## Competing interests

The authors declare the following competing interests: This work was part-funded by EditForce, Inc. B.G., M.I., T.N., Y.A., Y.S., and Y.Y. are inventors on a patent application related to DYW:KP proteins as well as other patents on PPR proteins. T.N. is a cofounder of EditForce, Inc. and all the authors are either employees or scientific advisor in the company.
