## [Peer Review File · Communications Biology]

Reviewers' comments:

Reviewer #1 (Remarks to the Author):

This paper describes the identification of PPR proteins, targeted base editors from U to C of RNA transcripts. The phenomena of U to C RNA editing in non-seed land plants are famous, and the PPR proteins were thought to be involved in it, but this is the first identification of some group of PPR proteins on the U-to-C editing. Although the proteins assayed in this paper were artificial and chimeric ones and they were expressed and assayed in heterologous organisms, demonstration of the activity was clear and satisfactory. Because this kind of proteins is awaited for the application biotechnology, & which will be used in in vitro and in human cells, the assays were rather suitable for the important insight of their usefulness in the application area. I enjoyed reading it well.

Reviewer #2 (Remarks to the Author):

In this manuscript, Ichinose et al. provide in vivo evidence that members of a specific subclade of PPR protein can perform U to C RNA editing. Their work also suggest that this activity could be used as a RNA editing tool in human cells, therefore paving the way for its potential application as a therapeutic tool in the future. The experiments are clever and nicely designed and the various claims from the authors are well supported. The in vivo editing experiments in both bacteria and human cells could appear redundant but the latter experiment was necessary to be convinced about the potential broader applications. Finally it should be highlighted that the manuscript provides the first near complete answer to a longstanding question in the plant organellar RNA editing field, the identity of the U to C editing enzyme. I however have minor concerns about the manuscript, about 1) the way some results are described and 2) the fact that the results have, in my mind, far reaching implications that are not discussed enough.

Although I am myself really familiar with the PPR protein world, I feel the first part of the results "Engineering a U to C editing protein" is confusing. First, why do the authors talk about the design of KP7 (line 59), just like it was the most important part of their work? This appears to be a minor construct that will then only be barely used compared to the other three ones (KP2, KP3 and KP). I feel all the writing between lines 56 to 68 should be made clearer. Basically, the design of the engineered PPR is complex enough to deserve a full figure dedicated to it. Here, useful information is spread out between Figure 1 and Figure 2A. Because of its importance, the design of both the (P1L1S1) PPR tracks and the C terminal (P2L2S2E1E2DYW) part should be explained clearly and a graphical view of the strategy would be very welcome. Understanding the strategy might also be made clearer if the model for the recognition of a RNA editing site by a PPR protein was explained beforehand. Explaining 1) the usual model for RNA editing site recognition, 2) the design of the P1L1S1 PPR track specific to the site chosen and 3) the design of the various C terminal part that will be fused to the P1L1S1 tracks seems more logical to me.

The other thing that could be more deeply discussed is the implications of the finding that the same editing factor can both perform U to C and C to U editing. Since the pioneering experiments that demonstrated in 1995 (Yu and Schuster; and Blanc and Araya) that C to U editing was a deamination reaction, it has been debated whether the enzymes were deaminases or transaminases. Because of the existence of the U to C reaction, the hypothesis of a transamination looked more parsimonious and the author's results clearly go into that direction. The caveat, however, is that the in vitro experiments mentioned above were performed using carefully dialyzed mitochondrial extracts, theoretically preventing the presence of the acceptor molecules necessary to transaminases. How could the authors reconcile their results with the biochemistry? Do they have any evidence, for example, that DYW KP proteins transfer amino groups on some of their own residues?

I really liked the experiments in human cells about the site specificity of the designer KP proteins. This is reminiscent and perfectly in line with tens of older experiments, either *in vivo*, *in vitro* or in organello that tried to dissect the *cis* acting elements necessary to the recognition of a RNA editing site. I understand this might not be the scope of the paper and could potentially be the topic of a full review, but one thing not really discussed in the manuscript is that it shows how good and robust the knowledge about plant organellar RNA editing has been, even before the discovery of the first PPR protein. All the work presented here is in line with what has been shown with more archaic methods, the perfect example being the Miyamoto paper (ref 21) briefly mentioned in the manuscript. I feel like these earlier results could be at least a little bit discussed here.

Other points include:

- Why didn't the authors also introduce mutations in KP2? What is the rationale for only using KP3 and KP7 (Figure 2)? KP2 seems to work better than KP3 and it is used in the work with human cells.
- Still in Figure 2, the recombinant proteins don't seem to display the same migration pattern on the SDS PAGE while they only marginally differ in their sequences (KP1 and KP7 for example migrate higher than the others). Can the authors comment on this?
- why didn't the authors perform the off target analysis in bacteria too?

Reviewer #3 (Remarks to the Author):

The work of Ichinose et al. describes the possibility to reconstitute in *E. coli* and human cells targeted U>C RNA base editing activity that is naturally observed in organelles of the three plant clades, hornworts, lycophytes and ferns. To achieve this goal, the authors used engineered synthetic PLS-PPR repeat guides fused to DYW:KP catalytic domains deriving from consensus amino acid sequences specifically found in these three plant species performing U>C RNA editing. While the possibility to use synthetic PLS PPR repeats to perform targeted C>U RNA editing in plant chloroplasts or bacteria has been reported (Royan et al, *Com Biol* 2021), the results presented in this manuscript are interesting and novel; synthetic DYW:KP domains add to the existing toolbox for genetic engineering, making possible the targeted editing of U>C in RNAs in living organisms.

The experimental results in the manuscript are globally convincing to support the uridine to cytidine editing function of the synthetic DYW:KP domains KP2, KP3, KP6 domain in bacteria and additionally in human cells for KP6. However, the conclusions provided in Figure 4 should be analyzed with more caution :

- The authors evaluated the specificity of the engineered L2S2E1-DYW:KP for its designated binding site by sequence mutagenesis (A, C, G, U) and measurement of the *in vivo* editing efficiency. The authors then conclude about the "specificity" of each repeat for one or another base without statistical tests. For example, they conclude that the L2 motif of KP2 and KP2 is specific to pyrimidine while the results show that it still recognizes purines but to a lesser extent. How significant is this difference between purines and pyrimidines? The same comment applies for all base preferences the authors assigned to each motif.

In addition, the weak point of the bacterial assay to test PPR/RNA recognition models using mutagenesis on the RNA sequence resides in limiting factors that can affect the interpretation of the results : 1) the level of expression of the RNA targets in bacteria and, 2) the introduction of RNA secondary structure by the mutations. These two could negatively impact the editing activity measured in bacteria. In the manuscript, the authors did not discuss these points.

- The RNA editing activity of the DYW-KP protein was reconstituted in bacteria and raises question about the in vitro activity of these proteins. Were the authors capable of reconstituting the cytidine deaminase activity in vitro using the purified recombinant proteins? If not, do the authors have a reasonable explanation for this negative result?

- line 202; the statement from the authors "KP6 is more prone to off-target because of the tolerance to mismatches in the predicted target sequence" is not convincing at all because the analysis of the in vivo off targets was not extensively performed to draw conclusions (i.e the full transcriptome in human cells was not analyzed) and second, the identified ZDHHC20 KP6's off-target only differs from its designated 16-nt target site by one nucleotide at the 5' end.

Reviewers' comments

Reviewer #1 (Remarks to the Author):

This paper describes the identification of PPR proteins, targeted base editors from U to C of RNA transcripts. The phenomena of U to C RNA editing in non-seed land plants are famous, and the PPR proteins were thought to be involved in it, but this is the first identification of some group of PPR proteins on the U-to-C editing. Although the proteins assayed in this paper were artificial and chimeric ones and they were expressed and assayed in heterologous organisms, demonstration of the activity was clear and satisfactory. Because this kind of proteins is awaited for the application biotechnology, & which will be used in in vitro and in human cells, the assays were rather suitable for the important insight of their usefulness in the application area. I enjoyed reading it well.

Reviewer #2 (Remarks to the Author):

In this manuscript, Ichinose et al. provide in vivo evidence that members of a specific subclade of PPR protein can perform U to C RNA editing. Their work also suggest that this activity could be used as a RNA editing tool in human cells, therefore paving the way for its potential application as a therapeutic tool in the future. The experiments are clever and nicely designed and the various claims from the authors are well supported. The in vivo editing experiments in both bacteria and human cells could appear redundant but the latter experiment was necessary to be convinced about the potential broader applications. Finally it should be highlighted that the manuscript provides the first near complete answer to a longstanding question in the plant organellar RNA editing field, the identity of the U to C editing enzyme. I however have minor concerns about the manuscript, about 1) the way some results are described and 2) the fact that the results have, in my mind, far reaching implications that are not discussed enough.

Although I am myself really familiar with the PPR protein world, I feel the first part of the results “Engineering a U to C editing protein” is confusing. First, why do the authors talk about the design of KP7 (line 59), just like it was the most important part of their work? This appears to be a minor construct that will then only be barely used compared to the other three ones (KP2, KP3 and KP. I feel all the writing between lines 56 to 68 should be made clearer. Basically, the design of the engineered PPR is complex enough to deserve a full figure dedicated to it. Here, useful information is spread out between Figure 1 and Figure 2A. Because of its importance, the design of both the (P1L1S1) PPR tracks and the C terminal (P2L2S2E1E2DYW) part should be explained clearly and a graphical view of the strategy would be very welcome. Understanding the strategy might also be made clearer if the model for the recognition of a RNA editing site by a PPR protein was explained beforehand. Explaining 1) the usual model for RNA editing site recognition, 2) the design of the P1L1S1 PPR track specific to the site chosen and 3) the design of the various C terminal part that will be fused to the P1L1S1 tracks seems more logical to me.

Response: *We have added a little more introduction on RNA recognition and re-arranged the text in the “Engineering a U-to-C editing protein” section as suggested (p. 3-5, lines 58-98). We merged Fig. 2a with a new figure explaining the design of the DYW:KP proteins (New Fig. 1).*

The other thing that could be more deeply discussed is the implications of the finding that the same editing factor can both perform U to C and C to U editing. Since the pioneering experiments that demonstrated in 1995 (Yu and Schuster; and Blanc and Araya) that C to U editing was a deamination reaction, it has been debated whether the enzymes were deaminases or transaminases. Because of the existence of the U to C reaction, the hypothesis of a transamination looked more parsimonious and the author’s results clearly go into that direction. The caveat, however, is that the in vitro experiments mentioned above were performed using carefully dialyzed mitochondrial extracts, theoretically preventing the presence of the acceptor molecules necessary to transaminases. How could the authors reconcile their results with the biochemistry? Do they have any evidence, for example, that DYW KP proteins transfer amino groups on some of their own residues?

Response:

We could not obtain any C-to-U or U-to-C editing activity in vitro making the study of biochemistry difficult. It is possible that the cofactor providing the amine group was missing in our experiment or the experimental conditions were not optimal. In the absence of experimental results, we can only speculate on the mechanism of the DYW domain. It is difficult to conceive that a same (KP6) or two highly similar enzymatic domains (DYW:PG and KP) catalyze completely different mechanisms. It is likely that the DYW:PG catalyze a more complex hydrolytic deamination involving maybe hydroxyl or amino groups on the amino acids backbone. However, we cannot rule out the hypothesis that the subtle modification in the amino acid sequence modify the structure to discriminate between an extra free water or ammonia.

We added more discussion on the reaction (p. 14 and lines 255-265). However, we didn't want to expand the discussion on the mechanism in the absence of additional results (particularly in vitro) as it is necessarily extremely speculative.

I really liked the experiments in human cells about the site specificity of the designer KP proteins. This is reminiscent and perfectly in line with tens of older experiments, either in vivo, in vitro or in organello that tried to dissect the cis acting elements necessary to the recognition of a RNA editing site. I understand this might not be the scope of the paper and could potentially be the topic of a full review, but one thing not really discussed in the manuscript is that it shows how good and robust the knowledge about plant organellar RNA editing has been, even before the discovery of the first PPR protein. All the work presented here is in line with what has been shown with more archaic methods, the perfect example being the Miyamoto paper (ref 21) briefly mentioned in the manuscript. I feel like these earlier results could be at least a little bit discussed here.

Response: *We added more discussion on this (p. 15 and lines 287-297).*

Other points include:

- Why didn't the authors also introduce mutations in KP2? What is the rationale for only using KP3 and KP7 (Figure 2)? KP2 seems to work better than KP3 and it is used in the work with human cells.

Response: We had not tested it for the first version of the manuscript because KP3 has a better editing activity than KP2 in human cells. Since then, we have tested it and have shown that this mutation surprisingly increases the U-to-C editing activity. We couldn't explain the reason with our current knowledge on the DYW domain.

We have added more text page 6 line 122 ("11% (KP2)") and page 8 lines 155-156 ("Interestingly, the editing efficiency of the KP2 HSE mutant increased by 10%, whereas no activity was detected with KP3 HSE confirming the important role of this amino acid in the function of the DYW domain.") and updated figures. 3c and 4c.

- Still in Figure 2, the recombinant proteins don't seem to display the same migration pattern on the SDS PAGE while they only marginally differ in their sequences (KP1 and KP7 for example migrate higher than the others). Can the authors comment on this?

Response: We could not identify a trivial cause explaining the difference in migration pattern (e.g. difference of charge) and any explanations would necessarily be speculative. However, we can hypothesize that the small differences in the amino acid sequence of KP1 and 7 could increase the thermal stability of the proteins, decreasing the binding of SDS molecules and thus, slowing the migration.

- why didn't the authors perform the off target analysis in bacteria too?

Response: We did not add the off-target analysis because we didn't find a strong sequence similarity between AtrpoA-U editing site and any sequences in the E. coli transcriptome. In the revised version of the manuscript, we

have included the off-target analysis by RNA-seq in human cells and chosen not to add any additional data on E. coli (p. 12-13 and lines 201-228, “off-target editing by DYW:KP proteins” section).

Reviewer #3 (Remarks to the Author):

The work of Ichinose et al. describes the possibility to reconstitute in E. coli and human cells targeted U>C RNA base editing activity that is naturally observed in organelles of the three plant clades, hornworts, lycophytes and ferns. To achieve this goal, the authors used engineered synthetic PLS-PPR repeat guides fused to DYW:KP catalytic domains deriving from consensus amino acid sequences specifically found in these three plant species performing U>C RNA editing. While the possibility to use synthetic PLS PPR repeats to perform targeted C>U RNA editing in plant chloroplasts or bacteria has been reported (Royan et al, Com Biol 2021), the results presented in this manuscript are interesting and novel; synthetic DYW:KP domains add to the existing toolbox for genetic engineering, making possible the targeted editing of U>C in RNAs in living organisms.

The experimental results in the manuscript are globally convincing to support the uridine to cytidine editing function of the synthetic DYW:KP domains KP2, KP3, KP6 domain in bacteria and additionally in human cells for KP6. However, the conclusions provided in Figure 4 should be analyzed with more caution :

- The authors evaluated the specificity of the engineered L2S2E1-DYW:KP for its designated binding site by sequence mutagenesis (A, C, G, U) and measurement of the in vivo editing efficiency. The authors then conclude about the “specificity” of each repeat for one or another base without statistical tests. For example, they conclude that the L2 motif of KP2 and KP2 is specific to pyrimidine while the results show that it still recognizes purines but to a lesser extent. How significant is this difference between purines and pyrimidines? The same comment applies for all base preferences the authors assigned to each motif.

Response: We have added information on significant differences in Fig. 5 (previously 4) and rephrased the text: we have replaced the ambiguous word “specific” by “preference” (“Designer DYW:KP proteins have a site preference” section, p. 9-10, lines 166-185).

In addition, the weak point of the bacterial assay to test PPR/RNA recognition models using mutagenesis on the RNA sequence resides in limiting factors that can affect the interpretation of the results : 1) the level of expression of the RNA targets in bacteria and, 2) the introduction of RNA secondary structure by the mutations. These two could negatively impact the editing activity measured in bacteria. In the manuscript, the authors did not discuss these points.

Response: A comment was added in the result section (p. 10, lines 186-188): “Overall, those results suggest that the three DYW:KP proteins have a site preference in the vicinity of the editing site if we exclude that mutations lead to unpredictable variabilities in gene expression and/or structural modifications.”

- The RNA editing activity of the DYW-KP protein was reconstituted in bacteria and raises question about the in vitro activity of these proteins. Were the authors capable of reconstituting the cytidine deaminase activity in vitro using the purified recombinant proteins? If not, do the authors have a reasonable explanation for this negative result?

Response: We could not detect any C-to-U or U-to-C editing activity in vitro. The U-to-C editing activity of KP2, 3 and 6 proteins in *E. coli* is low compared to the C-to-U editing activity of PpPPR56 and PpPPR65 (Oldenkott et al., 2019), two natural PPR proteins for which the editing activity was reconstituted in vitro (Hayes & Santibanez, 2020; Takenaka et al., 2021) suggesting that the design of the DYW:KP proteins is not yet optimal. We will test again the in vitro editing activity after optimizing the DYW domain. However, we cannot rule out that cofactors stabilizing the PLS-RNA complex in vitro or providing the amine group were missing in our experiment.

- line 202; the statement from the authors “KP6 is more prone to off-target because of the tolerance to mismatches in the predicted target sequence” is not convincing at all because the analysis of the in vivo off targets was not extensively performed to draw conclusions (i.e the full transcriptome in human cells was not analyzed) and second, the identified ZDHHC20 KP6’s off-target only differs from its designated 16-nt target site by one nucleotide at the 5’ end.

Response: We updated the “off-target editing by DYW-KP proteins” paragraph with an RNA-seq study of the off-target RNA-editing activity of KP2, 3 and 6 (p. 12-13 and lines 200-228, raw data:

<https://dataview.ncbi.nlm.nih.gov/object/PRJNA856069?reviewer=um9jq6h9nm0kieap8ak36goi57>)

Additional comment

After re-analysing the sequencing files for resubmission, we found that one nucleotide was missing in the plasmid used for KP2 / -1C value (Figure 5a). The updated value (0% to 9%) doesn’t change the interpretation of the result “the three DYW:KP proteins have a preference for purines and pyrimidines at positions -2 and -1 respectively”.

REVIEWERS' COMMENTS:

Reviewer #2 (Remarks to the Author):

I thank the authors for convincingly answering my comments and questions. For example, the new Figure 1 makes the understanding of the strategy easier. I only have a few minor comments left: While the off target editing analysis with RNA-Seq is powerful and convincing I'm disappointed that they completely removed from the paper their original approach, looking first for similar sequences in the genome and then checking for editing. How good is the overlap between their initial results and the RNA-Seq data? Checking rapidly it seems that their previous main candidate (ZDHHC20) indeed appears edited in the RNA-Seq data (genomic position 21391826 on the minus strand of the chromosome 13 if I'm not mistaken). What about the others? I feel like both approaches could be complementary.

Other points:

Legend of the figure 1 should be expanded. In its current state one has to read the material and method to fully understand the figure. The legend should (at least partially) describe the different steps and explain the formalism used. What do the black crosses mean for example? Etc.

Line 218: change homology by similarity

Reviewer #3 (Remarks to the Author):

In the revised manuscript and letter to reviewers, the authors have addressed all the comments I had raised by toning down some of their arguments and/or editing their conclusions with more explanations. The revised manuscript sounds more rigorous and convincing and therefore, I do not raise more concerns about its quality.

Reviewer comments

Reviewer #2 (Remarks to the Author):

I thank the authors for convincingly answering my comments and questions. For example, the new Figure 1 makes the understanding of the strategy easier. I only have a few minor comments left:

While the off target editing analysis with RNA-Seq is powerful and convincing I'm disappointed that they completely removed from the paper their original approach, looking first for similar sequences in the genome and then checking for editing. How good is the overlap between their initial results and the RNA-Seq data? Checking rapidly it seems that their previous main candidate (ZDHHC20) indeed appears edited in the RNA-Seq data (genomic position 21391826 on the minus strand of the chromosome 13 if I'm not mistaken). What about the others? I feel like both approaches could be complementary.

***Response:** We added a supplemental table (Supplementary Table S2) including the potential off-targets having one to three mismatches with AtrpoA editing site (gggenome.dbcls.jp; Human spliced RNA, RefSeq curated on GRCh38/hg38.p13, D3G 22.02). We have also added a little more text (see below and page 7 line 162~164 in the manuscript).*

KP6 created 98 U-to-C editing off-target sites in HEK293T. The PPR protein performed 33% of U-to-C editing on AtrpoA, whereas the average editing efficiency on the off-target editing sites was 16%, suggesting that the relatively high editing efficiency on AtrpoA is due to its location on the plasmid mRNA coding for the PPR protein. Among 255 potential off-target editing sites having less than four mismatches with AtrpoA, only three were edited (Supplementary Table 2). However, the sequence logo generated from an alignment of KP6 off-target sites shows a similarity of sequence with the AtrpoA editing site (Fig. 6b).

Other points:

Legend of the figure 1 should be expanded. In its current state one has to read the material and method to fully understand the figure. The legend should (at least partially) describe the different steps and explain the formalism used. What do the black crosses mean for example? Etc.

Response: As requested, we have extended the legend of Figure 1 (see below).

Figure 1. Schematic depiction of the workflow for designing the PLS and C-terminal domains of designer DYW:KP proteins. (a) The PILIS1 triplets used for the design of the PLS domain were isolated from land plant genomes and selected based on their location in the PLS domain: the ‘N-term’ triplets correspond to the first PILIS1 triplet in the PLS domain, the ‘Central’ triplets are preceded and followed by a PILIS1 triplet, and the ‘C-term’ triplets precede a P2L2S2 triplet. (b) The C-terminal domains were designed on the P2, L2, S2, E1 and E2 PPR-like motifs and DYW domains isolated in hornworts, lycophytes and ferns transcriptomes and Anthoceros angustus genome. After improving the motif database, a phylogenetic analysis was performed to isolate subclades of proteins. (c) After selecting the PPR motifs on their average length, a unique PLS domain composed of three PILIS1 triplets and seven C-terminal domains (one for each subclade of proteins identified in (b)) were designed based on consensus sequences. (d) The amino acids involved in the RNA recognition were mutated to recognize specifically AtrpoA. The DYW:KP proteins overexpressed in Rosetta 2 cells were tagged with an N-terminal thioredoxin (Trx) domain and His-tag (6His). The targeted editing site (AtrpoA) is localized downstream of the stop codon. Below each PPR motif, the two amino acids determining the target specificity are aligned with the AtrpoA editing site. U/C indicates the editing site. Arrows indicate the design flow.

Line 218: change homology by similarity

Response: As requested, “homology” has been replaced by “similarity”